# Handling Open-Vocabulary Constructs in Formalizing Specifications: Retrieval-Augmented Parsing with Expert Knowledge

**Mohammad Saqib Hasan**[*], **Sayontan Ghosh**[*], **Dhruv Verma**[*] , **Geoff Kuenning**[†], **Erez Zadok**[*], **Scott A. Smolka**[*] **& Niranjan Balasubramanian**[*]

[*] Department of Computer Science, Stony Brook University

[†] Department of Computer Science, Harvey Mudd College

```
{mdshasan, sagghosh, dhverma, ezk, sas, niranjan}
@cs.stonybrook.edu, geoff@cs.hmc.edu
```

## Abstract

We study the problem of *Open-Vocabulary Constructs* (OVCs)—ones not known beforehand—in the context of converting natural language (NL) specifications into formal languages (e.g., temporal logic or code). Models fare poorly on OVCs due to a lack of necessary knowledge *a priori*. In such situations, a domain expert can provide correct constructs at inference time based on their preferences or domain knowledge. Our goal is to effectively reuse this *inference-time*, expert-provided knowledge for future parses without retraining the model. We present *dynamic knowledge-augmented parsing* (DKAP), where in addition to the input sentence, the model receives (dynamically growing) expert knowledge as a key-value lexicon that associates NL phrases with correct OVC constructs. We propose ROLEX, a *retrieval-augmented parsing* approach that uses this lexicon. A retriever and a generator are trained to find and use the key-value store to produce the correct parse. A key challenge lies in curating data for this retrieval-augmented parser. We utilize synthetic data generation and the data augmentation techniques on annotated (NL sentence, FL statement) pairs to train the augmented parser. To improve training effectiveness, we propose multiple strategies to teach models to focus on the relevant subset of retrieved knowledge. Finally, we introduce a new evaluation paradigm modeled after the DKAP problem and simulate the scenario across three formalization tasks (NL2LTL, NL2Code, and NL2CMD). Our evaluations show that DKAP is a difficult challenge, and ROLEX helps improve the performance of baseline models by using dynamic expert knowledge effectively.[1]

## 1 Introduction

Formalizing natural language (NL) specifications is a fundamental NLP problem with applications in verification (Chen et al., 2023; He et al., 2022), code and proof generation (Yin et al., 2018; Wang et al., 2021; Wu et al., 2022). Modern parsers for these tasks are modeled using Seq2Seq (Hahn et al., 2021; Irving et al., 2016; He et al., 2022) models, which take the specification as input and output a formal representation as a sequence of n-grams.

A fundamental challenge in NL formalization is the presence of *Open-Vocabulary Constructs* (OVCs), phrases in the NL specification for which the formal-language construct is not known *a priori*. This is due to either poor training data coverage or a lack of the domain knowledge needed to anticipate all possible target constructs. OVC problems are common in formalisms used in verification, such as Linear Temporal Logic (LTL) (Rozier, 2011) and Signal Temporal Logic (STL) (He et al., 2022). This is due to phrases that cannot be

---

[1]Code and data are available at `https://github.com/StonyBrookNLP/rolex`

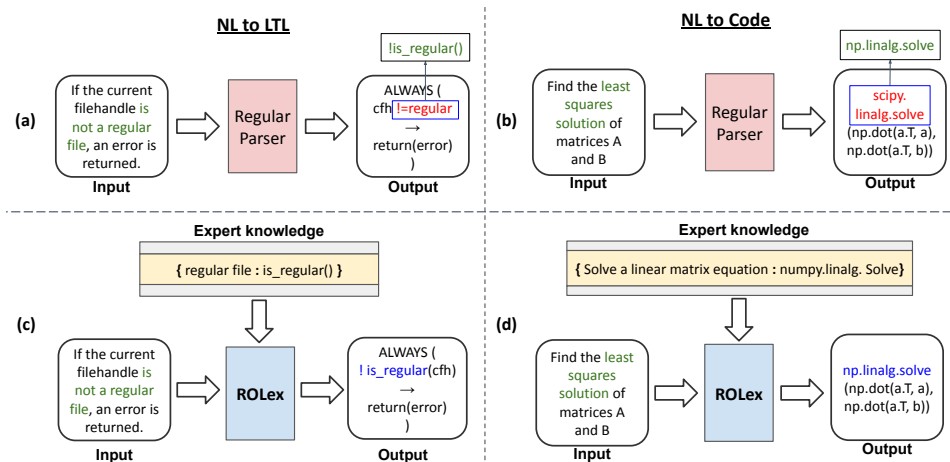

Figure 1: **ROLEX in practice**: We show two parsing tasks with OVCs: (i) natural language to linear temporal logic (NL to LTL) and (ii) natural language to code (NL to Code). As shown in **(a)** and **(b)**, a regular parser cannot generate the expected target constructs because they were not seen in the training data. The phrase IS NOT A REGULAR FILE in NL2LTL input should map to !IS_REGULAR(), but the model generates != REGULAR. Similarly for NL2Code, the model uses SCIPY.LINALG.SOLVE instead of NUMPY.LINALG.SOLVE as the library function for LEAST SQUARES SOLUTION. ROLEX augments models with the ability to use expert feedback to handle these OVC cases. The expert knowledge is designed as a key-value lexicon that maps NL phrases to their expected parses.

generalized to a formal construct without prior knowledge about the specific domain (*i.e.*, the software being verified) or expert preferences. Even domain experts cannot anticipate all necessary OVC constructs, since they would have to read (parse) each specification sentence, make decisions about how to represent the constructs and aggregate them into a list, making it a cumbersome process. Such constructs are, in effect, unknown to the model and the experts before they are first encountered when parsing the corresponding specifications. The NL-to-LTL portion of Figure 1 illustrates the use of OVCs with an example from Network File System specifications (Shepler et al., 2003). OVCs are also prevalent in regular program synthesis (Yin et al., 2018; Wu et al., 2022), as models may use functions from knowledge that does not meet the downstream runtime environment or user preference. The NL-to-Code portion of Figure 1 illustrates an example where a regular model might use a linear algebra implementation from the SCIPY library, whereas the user would prefer a NUMPY implementation due to its availability or runtime requirements.

OVCs present a challenge that is fundamentally different from previously studied out-of-distribution settings (Zhou et al., 2023; Yu et al., 2018; Mekala et al., 2023). OVCs are OOD problems in that the constructs in the test specifications are not seen by the model during training. However, in the settings we consider, the constructs are also not known or enumerable a priori: there is no knowledge available to use about the construct until the first encounter during inference. This means that we cannot apply techniques relying on documentation (Zhou et al., 2023) as a knowledge source for the construct or existing knowledge bases (Krishnamurthy & Mitchell, 2015) that provide the construct vocabulary (*e.g.*, predicates in tables). Other domain-adaptation techniques that use in-domain exemplars (Parvez et al., 2021; Pasupat et al., 2021) in retrieval-augmented-generation rely on models' generalization to generate the OVC construct given in-domain examples. This generalization, we argue, is a hard challenge for the types of OVC settings we consider, where deep domain expertise and user preferences impact construct design and choice.

To address the OVC issue, a model needs feedback from the user about the correct construct. In traditional feedback-based frameworks, user feedback is provided after every parse. However, in formalization tasks, we find that OVCs, while not known to the models during training, repeat multiple times in specification sentences. Therefore, OVC feedback from the user can be recorded and used in a later parse without any fine tuning. This presents a new knowledge-augmented parsing setup, which we call *dynamic knowledge-augmented parsing* (DKAP). In regular knowledge-augmented parsing, a static knowledge database (*e.g.*, Wikipedia, Github) is built before inference (Lewis et al., 2020; Yu, 2022). In DKAP, the knowledge base is dynamic, growing with each parse. DKAP presents a realistic scenario

for solving the OVC issue because (i) it is difficult to enumerate all possible OVCs before they are encountered, (ii) an expert can generate knowledge based on errors, and (iii) OVCs repeat across parses so knowledge can be reused.

Using DKAP, we propose ROLEX, a *retrieval-augmented generation* (RAG) framework, to help parse OVCs. ROLEX collects small amounts of feedback from the expert based on the errors made by the parser when translating OVCs. This feedback, designed as a key-value **lexicon**, represents knowledge from the expert for the model to use in translating an OVC in future parses. The key is a generalized NL phrase or description of an OVC; the value is the correct construct. A retriever module is trained to fetch the relevant set of expert lexicons from the feedback database during each parse, and a generator module is conditioned to use this information when parsing. When an already-seen OVC is encountered in a future parse, the previously stored knowledge is used to generate the correct parse. Figure 1 shows examples of how ROLEX will help resolve OVCs. We outline two mechanisms for training ROLEX: (i) using synthetic data when there is no training data and (ii) using data augmentation techniques when training data exists. We also propose several training methods that let the RAG modules learn the task better. Furthermore, we show how few-shot LLMs can also benefit when modeled via ROLEX.

We simulated ROLEX across several benchmarks from various semantic parsing domains: NL-to-Linear Temporal Logic (NL2LTL), NL-to-Bash commands (NL2CMD), and NL-to-Python code (NL2Code). The simulations were done in the DKAP scenario and in the presence of out-of-distribution constructs to emulate OVCs. We observed that ROLEX provides consistent performance gains over baseline seq2seq generation in both fine-tuned and few-shot settings. Further analysis shows how each module of ROLEX is designed to provide optimal performance.

In summary, this paper makes the following three key contributions: (1) introduces a new parsing setting called *dynamic knowledge-augmented parsing* (DKAP) that accounts for growing user-provided inference-time knowledge; (2) presents a retrieval-augmented generation model, ROLEX, and an effective training strategy to address the OVC problem in semantic parsing in the DKAP setting; (3) demonstrates the challenges of the DKAP problem and the effectiveness of the proposed solution (ROLEX) through empirical evaluation and analysis on three different formalization tasks.

## 2   Related Work

**Out-of-Distribution (OOD) and Zero-Shot Semantic Parsing:** Works like Spider (Yu et al., 2018) (Text2SQL) and Docprompting (Zhou et al., 2023) introduce out-of-distribution (OOD) semantic parsing benchmarks. Others develop models to handle OODs. For example, SeqZero (Yang et al., 2022) decomposes tasks into subtasks, (Shaw et al., 2021) combine model and grammar, Zheng & Lapata (2021) incorporate entity tags, and Bogin et al. (2021) train LMs to learn syntactic structure parsing. Zero/few-shot parsing is also popular owing to low data requirements. For instance, Zerotop (Mekala et al., 2023) decomposes the problem into subproblems and solves them as zero-shot QA tasks. In Wu et al. (2022), the authors generate theorem code from NL in zero-shot, while NL2TL (Chen et al., 2023) generates synthetic data using few-shot prompting for fine-tuning. Our work addresses the OVC issue where constructs are not known *a priori* and thus cannot be resolved before they are encountered. Hence, unlike OOD settings where there are domain shifts but constructs are known, the need to use constructs never seen during training makes our OVC setting much more challenging.

**Domain Adaptation in Parsing:** Researchers have proposed domain adaptation in semantic parsing via retrieval-augmented generation using exemplars (Pasupat et al., 2021) or meta-learning (Chen et al., 2020b). (Shi et al., 2022) address cross-lingual Text2SQL with RAG using source domain exemplars for solving target domain queries, while Chen et al. (2023), He et al. (2022), and Jia & Liang (2016) use synthetic data to learn parsing in the target domain. Our work also incorporates ideas of domain adaptation as synthetic data is created to train ROLEX. A key challenge specific to our work is that all the constructs required to generate the output are not known apriori. As such, even with synthetic data, generation

models are not exposed to all the possible constructs during training and need the ability to learn to use new constructs on-the-fly during the inference time.

**Retrieval-Augmented Generation (RAG) for Parsing:** One common RAG for parsing involves retrieving examples from training data (Pasupat et al., 2021; Gupta et al., 2022). Zemlyanskiy et al. (2022) introduce an additional step by doing another retrieval with the input and generated output and using that as the model input for the final parse. Wu et al. (2023) use cyclic training of RAG modules from unsupervised data based on confidence scores. Docprompting (Zhou et al., 2023) uses documentation to resolve queries about code, while REDCODER (Parvez et al., 2021) uses code-comment examples. Our solution is also based on RAG formulation and incorporates expert inputs at inference using a growing knowledge base of lexicons.

**OVCs in Semantic Parsing:** OVCs in parsing previously involved addressing generalization to new words, for instance, using character representations (El Boukkouri et al., 2020; Kawakami et al., 2017). However, with modern models, the problem domain has shifted to handling unknown labels and constructs before they are encountered during inference. Examples are Jiao et al. (2022), which introduces OVC arguments in role prediction problems, and Simig et al. (2022), which introduces the OVC problem in classification problems with a large number of labels. Some other works address OVC predicates in semantic parsing, such as Gardner & Krishnamurthy (2017), where freebase queries are resolved using knowledge base, and Krishnamurthy & Mitchell (2015), which proposes predicate matching post generation to handle OVCs. Das et al. (2021) use RAG to address OVC issues by retrieving similar training exemplars and test on out-of-distribution entities in a limited fashion. Dalvi Mishra et al. (2022) use a dynamic memory of feedback for open-domain QA. While similar to the above two in using a RAG-based solution to elicit user input, our work differs in terms of the target domain (semantic parsing for specifications versus QA for open-domains), and in its focus on a compact feedback format using key-value lexicons.

## 3 Handling OVCs with Dynamic Knowledge-Augmented Parsing

The *open-vocabulary construct* (OVC) problem refers to the presence of *a priori* unknown constructs from outside the model's vocabulary space. We motivate OVC in two settings: (i) formalizing specifications for verifying complex systems (*e.g.*, NL to Linear Temporal Logic (Brunello et al., 2019)) and (ii) text-to-code (*e.g.*, NL to Python code (Yin et al., 2018)).

In formal specifications, the NL constructs are often domain-specific (*i.e.*, tied to the specific software being verified) and thus a generic fixed list is not particularly useful. Also, for a given domain, the full set of needed constructs cannot be enumerated beforehand, since they can be identified and resolved (by experts) only after reading (parsing) the specification sentences. For example, in the Network File Systems domain, there are thousands of network protocol documents (called RFCs), each containing numerous specifications. Experts need to manually parse each such specification with full knowledge of all the possible constructs for building formal models. Experts in our own research group, people with years of experience in formal verification and file systems, struggled to parse these specifications due to the presence of OVCs that they did not know about a priori.

In text-to-code settings, a user might prefer to use a different implementation (*e.g.*, a preferred Python library) than what the model was trained on, implement their own method, or use a specific representation. Thus, unlike standard semantic parsing settings, these settings include open-vocabulary constructs, ones that are unseen or unknown during training, and are only resolved when a specs require construct encountered at test time.

### 3.1 Dynamic Knowledge-Augmented Parsing: Problem Definition

To address this challenge, we introduce the *Dynamic Knowledge-Augmented Parsing Problem* (DKAP), which includes the following changes to the standard semantic parsing setup: **(i) Dynamic growing knowledge:** When parsing a given test sentence, the models use a dynamically growing knowledge base, which records all knowledge provided by the user for the open vocabulary constructs encountered in previous sentences. **(ii) Reuse without**

| Domain | NL | LEX | FL |
|--------|-----|-----|-----|
| NL2LTL | If the current filehandle is not a regular file, an error will be returned to the client. | ⟨current filehandle → cfh⟩, ⟨A is a regular file → is_regular(A)⟩, ⟨A returned → return(A)⟩ | ALWAYS(( !(is_regular(cfh)) ) $\implies$ ( return(error) )) |
| NL2Code | get the dot product of two one-dimensional numpy arrays | ⟨ Dot product of two arrays ... →numpy.dot⟩ | np.dot(a[:, (None)], b[(None), :]) |

Table 1: Example of natural language specification (NL), relevant expert lexicons needed for parsing (LEX), and a translated formal language statement (FL) from NL to LTL and NL to Code domain.

**retraining:** To reduce the user's burden, we assume that they provide knowledge about a new construct the first time it is encountered. The models must learn to *reuse* this knowledge when the same construct reappears later. Appendix Section D empirically motivates the prevalence of OVCs in a real specification modeling scenario and their potential for reuse. Lastly, we want the models to be trained to use the knowledge as it becomes available, without having to retrain every time a new construct is handled.

Consider a stream of NL sentences $X = \langle x_1, \cdots, x_N \rangle$. In the standard parsing problem, a model produces an output $y_t$, conditioning only on the input $x_t$. In DKAP, when parsing the sentence $x_t$, the model is additionally given access to expert knowledge $K_{1:t-1}$, the accumulated knowledge for all open-vocabulary constructs that appeared in the input sentences $\langle x_1, \cdots, x_{t-1} \rangle$. For $x_t$, a model is thus expected to generate the output $y_t$, conditioning on both $x_t$ and the expert knowledge $K_{1:t-1}$.

### 3.2 Representing Dynamic Expert Knowledge Using Lexicons

We designed a simple and concise text-to-construct mapping to record feedback in the form of an *expert lexicon*. The keys in the lexicon are generic, idiomatic natural-language phrases; the values correspond to the correct (*i.e.*, desired) formal representations for the OVCs. Example lexicon entries from two domains, NL2LTL and NL2Code, are shown in Table 1. The NL2LTL example, from the Network File System (NFS) domain (Shepler et al., 2003), shows that the lexicon entry ⟨A IS A REGULAR FILE → IS_REGULAR(A)⟩ designating that any phrase resembling A IS A REGULAR FILE should be parsed to construct IS_REGULAR() with the argument being the entity A. Similarly, for NL2Code, the lexicon ⟨ Dot product of two arrays ... →numpy.dot⟩ indicates that the dot-product function from the NUMPY library should be used if the specification requires it. Formally, when parsing $x_t$, the available knowledge is given by $K_{1:t-1} = \left\{ (k_i, v_i) \right\}_{i=1}^{m_{t-1}}$, where $m_{t-1}$ is the total number of unique OVCs seen until $x_{t-1}$. Each key $k_i$ is a generalized natural-language phrase representing an OVC, and $v_i$ is its corresponding expert-provided formal representation.

## 4 Retrieval-Augmented Parsing with Expert Lexicons

To address the DKAP setting, we need models that can take an input NL specification $x_t$ and the current state of the expert lexicon database $K_{1:t-1}$ at $T = t$ and produce a semantic parse $y_t$ that includes the appropriate OVC constructs from $K_{1:t-1}$. A simple solution is to fit the entire knowledge $K_{1:t-1}$ as additional context to a generative model to produce the output parse. This can be problematic for two reasons. First, the growing knowledge base may exceed the model's size limit. For example, a common input limit for many Seq2Seq models is 512 tokens, which is, for example, exceeded after aggregating expert lexicons from only 25 NFS RFC specifications. Second, adding the entire $K_{1:t-1}$ may increase the *noise* (*i.e.*, irrelevant entries) in the input context for the generator. A retriever helps mitigate this problem by selecting the relevant parts of $K_{1:t-1}$ as input.

Therefore, we propose ROLEX, Retrieval-augmented generation parsing with expert-provided OVC lexicons, illustrated in Figure 2 using examples from the NFS RFC. We can fine-tune LMs such as T5 (Raffel et al., 2020) or Code-T5 (Wang et al., 2021), or few-shot prompt LLMs like ChatGPT (OpenAI, 2023a) and GPT-4 (OpenAI, 2023b)) for the generator

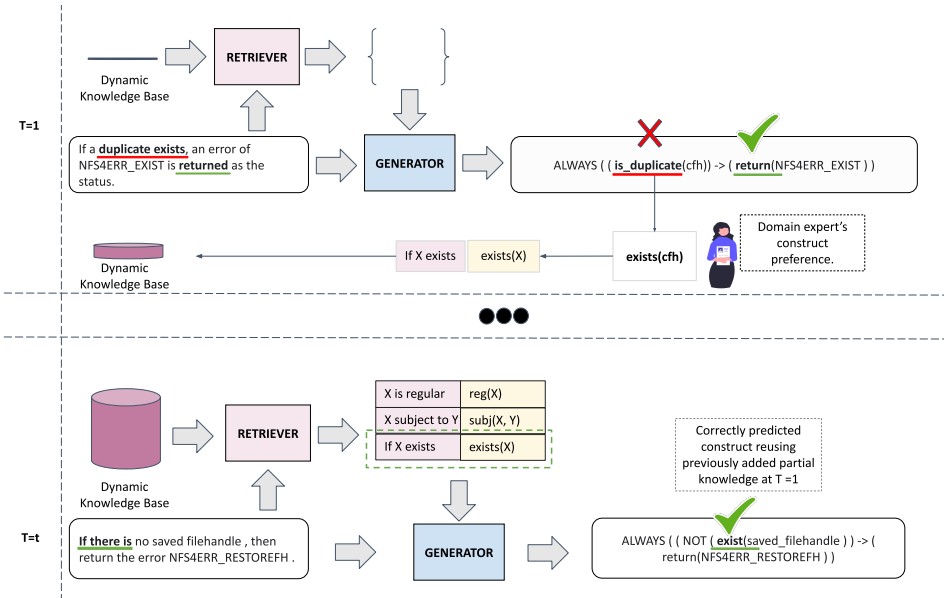

Figure 2: **Dynamic knowledge-augmented parsing using ROLEX**: Our RAG framework for inference-time knowledge-augmented parsing through expert lexicons. We start at $T = 1$ with an empty lexicon. The generator thus parses the first NL specification with empty retrieved knowledge to produce a formal statement. This generates an error (highlighted in red): "*duplicate exists*" is incorrectly parsed as IS_DUPLICATE(), whereas the domain expert prefers "*exists(X)*". This partial-knowledge entry is added to the dynamic knowledge base and gets reused appropriately at a later time step $T = t$, as shown in the lower block.

module, and use dense retriever models (Gao et al., 2021; Karpukhin et al., 2020) for the retriever. The main challenge lies in obtaining training data. To address this, we use synthetic data generation and data-augmentation techniques. To develop the retriever and generator modules, we experimented with several methods, explained next.

## 4.1 Dataset Creation and Augmentation

To train the generator, we need training instances of the form $(x, K, y)$; to train the retriever, we need instances of the form $(x, K, K_y)$, where $K_y \subseteq K$ is the subset of relevant key/value pairs for the OVCs that appear in the output $y$ for instance $x$. Here we describe our approach for generating and augmenting such training data automatically.

**Synthetic Data Generation:** In some semantic parsing use cases, such as formalizing RFC specifications, manually curating $(x, y)$ pairs and lexicons for OVCs can be difficult. However, due to the restricted nature of the parses, we can curate synthetic data via grammar-based generation (He et al., 2022). We do this by developing a synthetic grammar $\mathcal{G}$ adapted to the target domain. For example, we have shown that such a grammar can be used to model LTL statements generated from the NFS domain. Details of the grammar are provided in the Appendix Section A.

**Data Augmentation Setting:** If training data with $(x, y)$ pairs exist, we can augment them with the gold lexicon $K$ for each pair. We do this by first defining the OVCs and their formal representations, and then using a suitable pre-defined knowledge source to obtain an idiomatic natural-language description of the OVCs. For example, the first few words from Python documentation serve as OVC descriptions in the NL2Code domain. In this way, we construct a comprehensive lexicon knowledge base and then use it to enumerate each $(x, y)$ pair with the relevant $K$.

## 4.2 Developing the RAG Modules

**Retrieval Module:** We use sentence transformers (Reimers & Gurevych, 2019) as our dense retrievers in two ways: off-the-shelf and fine-tuning. For *off-the-shelf* usage, we plug in a

pre-trained retriever. However, this approach does not work well in formalization settings because most retrievers are fine-tuned on general NL tasks, and each parsing domain has its unique language and set of sentences to model. Hence, we opt for *fine-tuning the retriever* using our curated $(x, K, y)$ data pairs. Given $K = \{(k_i, v_i)\}_{i=1}^{m}$, where $k_i$ is the natural-language grounding phrase and $v_i$ is the OVC's formal representation, we create $(x, k_i)_{i=1}^{m}$ tuples for each $x$. This tuple store is then used to fine-tune a retriever $\mathcal{R}$ using contrastive learning (Henderson et al., 2017) with in-batch negatives (Karpukhin et al., 2020; Chen et al., 2020a) (see the Appendix Section B, for the training objective). The retriever $\mathcal{R}$ is fine-tuned to include all $K$ gold lexicon entries in its top $n$ ranked lexicon entries.

**Generator Module:** The Generator $\mathcal{G}$ is a language model that should generate the semantic parse $y$ based on $(x, K^*)$, where $K^*$ is the list of top $n$ ranked lexicon entries retrieved by $\mathcal{R}$ from the full lexicon. Ideally, the generator should be able to identify $K_y$, the relevant lexicon entries in $K^*$ and then use them appropriately to produce the correct semantic parse $y$. However, in practice, there can be three types of issues that arise when training generators with noisy retrieved lexicon entries. The generator might: (i) entirely ignore $K^*$ during generation, (ii) use non-relevant entries in $K^*$, resulting in incorrect parses, and (iii) fail to identify or use *all* of the relevant lexicon entries. To mitigate these issues, we propose four generator training schemes: **(1)** BASIC: $\mathcal{G}$ is trained by generating $y$ conditioned on $x \oplus K^*$. **(2)** EXTRA SUPERVISION: From each $(x, K^*, y)$ training pair, we derive two pairs: $(x, K^*, y)$ and $(x, K_y, y)$. This causes $\mathcal{G}$ to observe $y$ during training twice—once conditioned on $x \oplus K_y$ and another time on $x \oplus K^*$. The idea is to provide an extra supervision signal to $\mathcal{G}$ on what are correct lexicons. **(3)** MULTI-TASK LEARNING: $\mathcal{G}$ is trained to generate $k_y \oplus y$ conditioned on $x \oplus K^*$. The motivation lies in workings of language models: they generate the next token based on the input and the prior generation. **(4)** TRANSFER LEARNING: $\mathcal{G}$ is trained first to generate $k_y$ conditioned on $x \oplus K^*$. Then model is further fine-tuned as in scenario (1). Intuition is that relevant lexicon extraction helps $\mathcal{G}$ learn to parse better.

## 5 Evaluation Setup

Our goal is to evaluate ROLEX for addressing OVC issues in the Dynamic Knowledge-Augmented Parsing (DKAP) setting. We introduce a new evaluation protocol that differs from the standard evaluation for RAG: (i) the knowledge base grows as we parse more specifications and (ii) an expert adds new knowledge to correct OVC errors after each parse.

**DKAP Evaluation Protocol:** Our evaluation protocol mimics dynamic (growing) knowledge of OVCs at test time by simulating the addition of knowledge about a new OVC after each parse. In this way, subsequent parses can use this accumulated knowledge. Formally, given task $\mathcal{T}$ and possible OVCs $O$, we create train and test sets, $\mathcal{D}_{train}$ and $\mathcal{D}_{test}$ as per methods in Section 4.1. Each $\mathcal{D}$ consists of tuples $(x_t, K_t, y_t)$ where $x_t$ is the NL, $y_t$ is the FL and $K_t$ are the gold lexicons. ROLEX is trained on $\mathcal{D}_{train}$ and then evaluated with an initially empty dynamic knowledge base, $\mathcal{KB}$. At time $t$, the retriever retrieves relevant lexicons $K^*$ from $\mathcal{KB}$, and $x \oplus K^*$ is given as input to generate $y'$. Comparing $y'$ and $y$, we extract and add a set of new lexicons $K_{new} \in K_t$, s.t. $K_{new} \cap \mathcal{KB} = \varnothing$ to $\mathcal{KB}$ so that they can be retrieved for parsing at time $t + 1$.

### 5.1 Task and Dataset Details

We evaluated ROLEX across three formalization tasks. Our evaluation sets were created with constructs in the target output that were not present in the training sets, thus simulating out-of-vocabulary constructs (OVCs). The first task, **NL2LTL**, was to parse NFS specs into Linear Temporal Logic (LTL); the OVCs are function predicates (verbs) and a subset of variables (nouns). Due to lack of training data, we opted for grammar-based synthetic data generation (outlined in Section 4.1). Details about the grammar are given in Appendix Section A. Our test set consisted of 100 NFS statements taken from SpecNFS (Ghosh et al., 2022) and annotated to LTL by experts. Our second task is **NL2Code**, which involves formalizing specs to Python code, and the OVCS are function names. We use the CoNaLa dataset from Docprompting as the test set that contains unseen functions to simulate OVCs. We augment the data with lexicons using the database of Python documents provided with

| Model | NL2LTL | | | | NL2Code | | | | NL2CMD | | | |
|---|---|---|---|---|---|---|---|---|---|---|---|---|
| | All | OVCs | | | All | OVCs | | | All | OVCs | | |
| | bleu | Prec. | Rec. | F1 | bleu | Prec. | Rec. | F1 | bleu | Prec. | Rec. | F1 |
| T5-B | 27.94 | 39.35 | 43.42 | 41.28 | 18.51 | 4.89 | 4.84 | 4.86 | 13.17 | 4.96 | 7.22 | 5.88 |
| +ROLEX | 53.28 | 68.67 | 55.75 | 61.54 | 23.02 | 18.85 | 20.50 | 19.64 | 15.44 | 25.47 | 37.39 | 30.30 |
| CT5-S | 30.59 | 34.83 | 24.75 | 28.94 | 16.90 | 5.68 | 5.13 | 5.39 | 18.10 | 3.47 | 5.39 | 4.22 |
| +ROLEX | 42.95 | 62.42 | 43.33 | 51.15 | 25.81 | 15.93 | 16.17 | 16.05 | 22.27 | 29.10 | 37.18 | 32.65 |
| CT5-B | 25.15 | 21.33 | 14.08 | 16.97 | 29.30 | 14.98 | 14.30 | 14.63 | 26.04 | 2.46 | 4.31 | 3.13 |
| +ROLEX | 54.01 | 63.83 | 48.00 | 54.80 | 31.81 | 17.35 | 18.85 | 18.07 | 27.11 | 20.83 | 32.00 | 25.24 |

Table 2: Performance gains in fine-tuned models across three different benchmark tasks due to ROLEX. Bleu in All was evaluated for the complete model output expression, whereas OVC's F1, Precision & Recall were evaluated only for the subset of OVC constructs in the target expression.

the dataset. Our final task, **NL2CMD**, involves converting statements to bash commands where the OVCs are bash functions. Here, we again use TLDR dataset from Docprompting as the test set that contains OOD commands. To create the lexicon, we utilize tldr[2] pages to form lexicon keys. Further details and examples from each dataset, *along with information on how the lexicons were designed*, are listed in Appendix Section C.

### 5.2 ROLEX Benchmarking Setup

We used the different (small, base, large) variants of the BGE sentence transformer from FlagEmbedding (Xiao et al., 2023) as our retriever. For the generator, we used T5 and Code-T5 for the fine-tuned setting and ChatGPT and GPT-4 for few-shot. For evaluation, we calculate BLEU (Papineni et al., 2002) to measure quality of whole parse and precision, recall, and F1-score of the OVCs to measure performance on OVCs. We also calculate the gains ROLEX achieves using reusable feedback compared to traditional feedback by measuring reduction in error. Further details are given in Appendix Section E.

## 6 Results & Analysis

Our experimental evaluation setup is aimed at answering the following questions: (i) How effective is ROLEX in leveraging expert knowledge to address OVCs? (Tables 2 and 3(a)) (ii) What is the effect of retriever performance on the overall downstream task performance? (Table 3(b)). (iii) What is the variation in the overall OVC task performance for different generator training approaches? (Table 4(a)). (iv) What is the effect of the format of expert knowledge on the OVC task? (Table 4(b)). Note all our evaluations are conducted on testbeds containing OVCs unseen during training to reflect the actual nature of the setup.

### 6.1 Results with fine-tuned LLMs

Table 2 reports ROLEX's performance when different pre-trained LMs are fine-tuned as generators for the three different OVC tasks. For each model, we compared a baseline generator that parses FL from NL without additional context against the corresponding ROLEX generator. All generators were trained using the TRANSFER LEARNING method, where we first train the model to extract lexicon followed by training it on the downstream task (see section 4.2). For the retriever, we used the large variant of the BGE model, which was fine-tuned for the lexicon retrieval. Without dynamic knowledge, models fare poorly on the OVCs across all three tasks. The results also show that ROLEX yields substantial improvements on the OVCs as well as on the overall generation, when compared to the baseline that does not use any additional knowledge. In NL2LTL, for example, the gains range from 20% to 38% in OVC F1. These improvements show that ROLEX uses the dynamic knowledge during inference to address the OVC issue.

---

[2]https://tldr.sh/

| Model | All | OVCs | | |
|---|---|---|---|---|
| | BLEU | Prec. | Rec. | F1 |
| ChatGPT | 42.23 | 56.70 | 38.14 | 45.61 |
| +bm25 | 51.27 | 67.18 | 50.00 | 57.33 |
| +ROLEX | 53.03 | 71.43 | 53.26 | 61.02 |
| GPT-4 | 46.98 | 61.86 | 39.60 | 48.29 |
| +bm25 | 62.10 | 79.64 | 70.02 | 74.52 |
| +ROLEX | 66.38 | 78.87 | 75.43 | 77.11 |

(a) Few-shot model performance

| Retriever | Retr. | OVCs | | |
|---|---|---|---|---|
| | R@10 | Prec. | Rec. | F1 |
| Off-the-shelf | | | | |
| -bm25 | 2.59 | 10.88 | 10.44 | 10.66 |
| -small | 17.91 | 13.64 | 13.59 | 13.62 |
| -large | 18.81 | 14.20 | 14.62 | 14.40 |
| Fine-tuned | | | | |
| -small | 19.37 | 15.30 | 15.33 | 15.31 |
| -large | 20.38 | 17.35 | 18.85 | 18.07 |

(b) Effect of retriever performance on downstream task

Table 3: **(a)** Few-shot model performance on the NL2LTL task. Augmenting with ROLEX showed consistent performance improvements across all of the metrics. **(b)** Effect of retriever performance (top@k recall) on the NL2Code task with the best-performing model Code-T5-base.

| Training Strategies | All | OVCs | | |
|---|---|---|---|---|
| | BLEU | Prec. | Rec. | F1 |
| BASIC (1) | 29.94 | 16.56 | 17.30 | 16.92 |
| EXTRA SUP. (2) | 27.69 | 15.77 | 16.80 | 16.27 |
| MULTITASK (3) | 26.53 | 18.72 | 21.21 | 19.89 |
| TRANSFER (4) | 31.81 | 17.35 | 18.85 | 18.07 |

(a) Effect of training strategies

| Feedback | All | OVCs | | |
|---|---|---|---|---|
| | BLEU | Prec. | Rec. | F1 |
| EXMP | 33.33 | 14.98 | 15.96 | 15.46 |
| ROLEX | | | | |
| -DOCS | 29.54 | 15.69 | 15.30 | 15.49 |
| -LEX | 31.81 | 17.35 | 18.85 | 18.07 |

(b) Evaluating different feedback mechanisms

Table 4: **(a)** Effect of generator training strategy on the downstream task of NL2Code, with BGE-large as the retriever. **(b)** Evaluating different feedback mechanisms for RAG.

## 6.2 Results with Few-Shot LLMs

Table 3(a) shows the results of few-shot LLMs as generators in three settings: (i) no external knowledge is provided, (ii) bm25 retriever (Robertson et al., 2009) is used to retrieve external knowledge, and (iii) fine-tuned bge-large retriever (ROLEX) is used to retrieve the external knowledge. We provide three few-shot examples. We observe consistent gains in both OVC F1 and the overall target generation BLEU for ROLEX against baseline and when using an bm25 retriever. GPT-4 + ROLEX gave the best performance on the dataset, even against fine-tuned versions. However, we observe that the best fine-tuned ROLEX (T5-base) from Table 2 performed better than both few-shot LLM baselines. While adding ROLex to GPT-4 performs the best, we find that T5-base with ROLex has similar performance to that of ChatGPT with ROLex. These results suggest the potential of smaller models fine-tuned with ROLEX when compared to significantly larger models.

## 6.3 Analysis

We analyzed the different components of ROLEX to justify our design choices. All analyses were conducted on the NL2Code benchmark using the Code-T5 base as the ROLEX generator. Finally, we studied the potential gains of ROLEX for each benchmark.

**Retriever Analysis**: Table 3(b) reports the top-10 recall (R@10) performance of different retrievers, and the precision, recall and F1 for the OVCs in the downstream task. We tried bm25, BGE-small and large retrievers, both with and without fine-tuning. We observe that dense retrievers (BGE) outperform BM25 by at least 15% on retriever recall. Fine-tuning and increasing model size also lead to improved performance. We can also see that improved retriever performance translates to better downstream task scores. As the best retriever performance is quite low, improving retriever performance is a direction worth pursing to do well on the OVC parsing task. Further details are given in Appendix Section F.

**Generator Training Strategies**: We saw that retrieving the relevant lexicon is challenging, so we want generators that can generate the target based on the noisy retrieved knowledge. For this, we evaluated four different generator training strategies: **BASIC** Directly using the retrieved knowledge to train the generator. **EXTRA SUP.** Using extra supervision by mixing in instances with only the gold knowledge. **MULTITASK** Getting the model to focus on

the relevant knowledge via *multi-task learning*. **TRANSFER** Getting the model to focus on the relevant knowledge via a *transfer learning* approach. Model performance on these four approaches is reported in Table 4(a). From the table we observe that (**APPROACH 1, 2**) are not up to the mark, whereas multi-task and task-transfer learning show substantial improvement over the first two. Based on the experimental results we choose the transfer-learning approach as our go-to generator training approach as it provides the best balance between the OVC F1-score and overall parse quality.

**How useful is the lexicon based encoding of knowledge?**: In Table 4, we analyze different forms of retrieval context to help models overcome OVC issues. We trained generators to use exemplars (Parvez et al., 2021) (EXEMPLARS) and documentation (Zhou et al., 2023), the lookup key in our lexicon, (DOCS); (see Appendix Section G for details). The results show that while exemplars have better overall BLEU, ROLEX with lexicons (LEX) gets the best performance on OVCs. This is because the (spec, FL) exemplars only provide generic signals about good parses and not specific information about how to represent the OVCs in the test sentence. Also, lexicons are better than just documentation as context, in part due to the succinct description of the relevant information needed to connect the NL to the corresponding FL construct.

**Estimated Reduction in Expert Effort** In a standard feedback system the expert has to read every output and correct each erroneous OVC. ROLEX aims to reduce this burden on the expert by making the feedback reusable. We calculated the maximum possible percentage gains in saving human effort (treating reading and correcting entries as unit-cost actions) for each of the fine-tuned benchmarks and for the few-shot scenario. We observe 24.3% savings for NL2TL, 5.6% for NL2Code, 11.9% for NL2CMD, and 32.9% for few-shot learning in NL2LTL. These results indicate that dynamic user feedback is a fruitful direction worth pursuing in settings where the OVC problem is prevalent. Please see Appendix Section H for more details on these results.

**Error Analysis of OVCs**: For the NL2LTL task, we analyzed the OVC errors for 25 T–base generations with the lowest BLEU scores. We observed that 23.3% of the errors were due to relevant expert knowledge in the retrieved lexicons not being used for generation. About 23.3% of the errors can be attributed to incorrect lexicon from the set being used. Almost 46.7% incorrectly generated OVCs were due to generator hallucination. For NL2Code we find that 56% are cases where relevant retrieved lexicons are not used, 28% are cases where relevant lexicons are not retrieved and 16% where an incorrect lexicon is used. This analysis shows that ROLEX's ability to use lexicons and not hallucinate are key issues to address for better performance. Further details are given in Appendix Section I.

# 7 Conclusion & Future Work

Formalizing specifications requires expert feedback because of OVCs not resolvable *a priori* without domain knowledge. In this work, we propose a new dynamic knowledge-augmented parsing setup and a RAG model to tackle the OVC problem. We show how our approach addresses the OVC issue by accumulating small, reusable expert feedback as a lexicon. Furthermore, we analyzed different training schemes to develop our model. Evaluations on multiple benchmarks highlighted the challenge of our new parsing setting and demonstrated our approach's ability to address it. However, there is further room for improvement, especially in developing models that can better utilize expert knowledge, and are generalizable to more domains both in terms of generation and retrieval. We also hope our new problem setup spurs future research not only on improved modeling but also on user-centric evaluations of feedback-based parsers.

**Acknowledgements**

We thank the anonymous reviewers of COLM for their feedback which helped improve the paper. This work was possible thanks to the NSF award CCF-1918225.

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

# Appendix

## A  Grammar for synthetic generation

Figure 3 shows the custom grammar designed to model and generate the synthetic NL2LTL dataset. We can generate example statements in the language defined by this synthetic CFG by stochastically chaining the production rules beginning with the start symbol. Since the natural-language phrases are also attached to the grammar, the process simultaneously yields the corresponding NL statement $x$ for the generated formal statement $y$. We can create a large number of such $(x, y)$ pairs by sampling from this grammar. To simulate the expert-provided knowledge, we augment each pair with a knowledge set $K$ that contains a random collection of key/value pairs as distractors, and also the relevant key/value pairs that correspond to specific OVCs appearing in the output $y$. We can thus control/vary how the distractors are created.

| | | | |
|---|---|---|---|
| 1. | <start> | $\rightarrow$ | G(<phrase>) |
| 2. | | $\rightarrow$ | G(<phrase> $\implies$ <phrase>) |
| 3. | | $\rightarrow$ | G(<phrase> $\implies$ <phrase> $\vee$ $\neg$ <phrase> $\implies$ <phrase>) |
| 4. | <phrase> | $\rightarrow$ | <$\alpha$> |
| 5. | | $\rightarrow$ | <$\alpha$> <con> <$\alpha$> |
| 6. | | $\rightarrow$ | <$\alpha$> <con> <$\alpha$> <con> <$\alpha$> |
| 7. | <$\alpha$> | $\rightarrow$ | <pred> |
| 8. | | $\rightarrow$ | <b_pred> |
| 9. | | $\rightarrow$ | <act_pred> |
| 10. | <pred> | $\rightarrow$ | $x \mid x \circ x \mid x_1 \circ x_2$ |
| 11. | | $\rightarrow$ | $\neg x \mid \neg(x \circ x) \mid \neg(x_1 \circ x_2)$ |
| 12. | <b_pred> | $\rightarrow$ | $x \geq u_1 \wedge x \leq u_2$ |
| 13. | | $\rightarrow$ | $x_1 \geq x_2 \wedge x_1 \leq x_3$ |
| 14. | | $\rightarrow$ | $x < u_1 \vee x > u_2$ |
| 15. | | $\rightarrow$ | $x_1 < x_2 \vee x_1 > x_3$ |
| 16. | <act_pred> | $\rightarrow$ | verb(E) WITH E.<pred>$_1$ ... E.<pred>$_4$ |
| 17. | <con> | $\rightarrow$ | $\vee \mid \wedge \mid X$ |
| 18. | $x$ | $\rightarrow$ | var |
| 19. | | $\rightarrow$ | get_var$_1$(var$_2$) |
| 20. | $var$ | $\rightarrow$ | $variable$ |
| 21. | | $\rightarrow$ | $noun$ |

Figure 3: The grammar used to generate the synthetic dataset. Many constructs from general LTL have been reduced for use in our problem domain. Constructs created for our domain are highlighted in red. The verb predicate consists of all verbs in the SpecNFS dataset whose operations cannot be defined by any Boolean operation and thus require special functions. get_VAR consists of constructs that capture the "X of Y" statements in the SpecNFS and NFS RFC domains.

## B  Retriever Loss Function

From Xiao et al. (2023), given $p$ and $q$ are the paired texts, $q' \in Q'$ is a negative example and $\tau$ is the temperature and $e$ represents encodings.

$$L = min. \sum_{(p,q)} -\log \frac{\exp^{\langle e_p, e_q \rangle / \tau}}{\exp^{\langle e_p, e_q \rangle / \tau} + \sum_{Q'} \exp^{\langle e_p, e_{q'} \rangle / \tau}} \tag{1}$$

## C   Task Dataset Details

The details about each dataset are given below:

**NL2LTL:** The task was to parse specifications of NFS operations into Linear Temporal Logic (LTL) statements. Obtaining training data is expensive, so we used the synthetic data generation process outlined in Section 4.1. We describe the grammar (Figure 3) and generation process above, in Section A. The OVCs in this task were names of function predicates (verbs) and a subset of variables (nouns), as indicated in red in our grammar. We evaluated the model (trained on synthetic data) on a test set of 100 real NFS requirement statements sampled from Ghosh et al. (2022) and annotated with their LTL expression by a group of experts (three university professors and two graduate students).

**NL2Code:** The task was to parse natural-language specifications to Python code. Here, Python function names were the OVCs. We used the CoNaLa training and test cases from Zhou et al. (2023) for data augmentation. To add expert knowledge, we used the documentation database provided with the dataset. The expert lexicon was designed such that the keys were the first 200 characters from the documentation of the corresponding function, and the values were the function names. We used the pre-defined test sets, as they contained functions unseen during training, thereby simulating OVCs.

**NL2CMD:** The task was to parse natural-language queries into Bash statements. Command names such as cat, ls, etc., were used to simulate OVCs. We used the TLDR dataset from Zhou et al. (2023) for our experiments. As training data is available, we augmented it with expert lexicons to train ROLEX. The lexicon key was a concise natural-language statement describing the command, created by taking the first line from the tldr page for the command. The lexicon's value was the command name. We tested with the predefined test sets from the TLDR dataset, as it contains the OOD command names needed to simulate OVC.

Table 5 shows examples of natural-language input, formal-language output, and the gold expert-knowledge lexicon from each task dataset that was used in our simulations. Table 6 shows further details about the number of data rows available/used for training and testing ROLEX.

| **TASK 1** - NL2LTL |
|---|
| **NL (NFS-RFC specification)** : If the current filehandle is not a regular file, an error will be returned to the client. |
| **Expert Knowledge** : ⟨current filehandle → cfh⟩, ⟨A is a regular file → is_regular(A)⟩, ⟨A returned → return(A)⟩ |
| **FL (Linear Temporal Logic)** : ALWAYS(( !(is_regular(cfh)) ) ⟹ ( return(error) ) ) |
| **TASK 2** - NL2CMD |
| **NL (Instruction/Query)** : Delete a shared memory segment by id |
| **Expert Knowledge** : ⟨Delete IPC (Inter-process Communication) resources.→ipcrm⟩ |
| **FL (Bash statement)** : ipcrm -shmem-id shmem_id |
| **TASK 3** - NL2Code |
| **NL (Instruction/Query)** : divide the values with same keys of two dictionary 'd1' and 'd2' |
| **Expert Knowledge** : ⟨ class float([x]) Return a floating point number constructed from a number or string x. If the argument is a string, it should contain a decimal number, optionally preceded by a sign, and optionally e ... → python.library.functions#float ⟩ |
| **Python** : {k: (float(d2[k]) / d1[k]) for k in d2} |

Table 5: **Evaluation Task Examples**: This table illustrates the NL input, target FL output, task example, and expert knowledge from the datasets for each task.

## D   Reusability in Semantic Parsing

Figure 4 shows the potential for reusing expert knowledge lexicons in an inference-time knowledge setting in the NFS RFC domain. The diagram highlights the repetition of knowledge and the potential of reusability of knowledge about parsing OVCS.

| Dataset | Retriever Train Set | Generator Train Set | Test Set |
|---|---|---|---|
| NFS RFC | 120k | 120k | 100 |
| TLDR | 8260 | 8260 | 928 |
| Conala | 2135 | 2135 | 543 |

Table 6: Size of the different training and test sets for each the datasets used in evaluating the tasks.

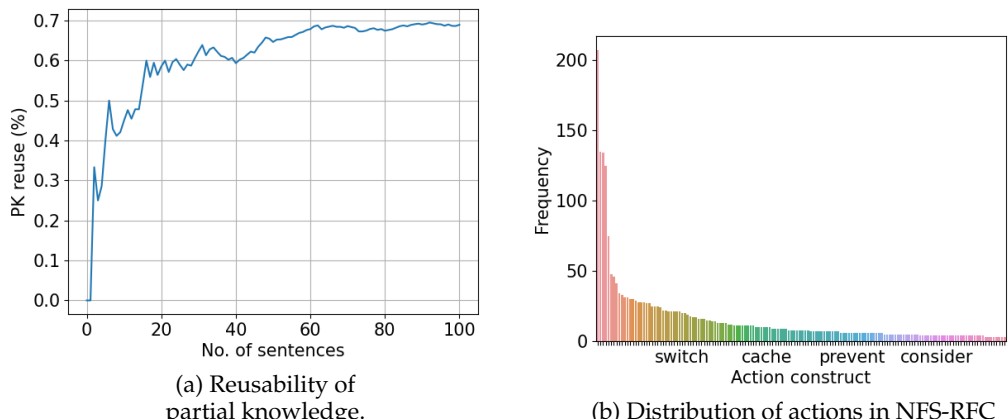

(a) Reusability of partial knowledge.

(b) Distribution of actions in NFS-RFC

Figure 4: **Observing reusability** : Analyses of the NFS specifications provides empirical support for the importance of modeling semantic parsing in an inference-time expert-knowledge setting. The left half (a) shows the percentage of reuse that can happen when we collect the formal constructs in the expert knowledge base. As we collect constructs from more sentences, the potential for reuse grows quickly. The right half (b) shows the frequency of action-denoting words (possible OVCs) in the specification sentences. There is a long tail of words that occur multiple times. Knowledge learned about these once will likely be useful multiple times later.

## E  Evaluation Setup and Metrics

Further details of the evaluation setup are given below:

**Retriever Training**: We utilize the small, base, and large variants of FlagEmbedding models (Xiao et al., 2023), which are sentence transformers (Reimers & Gurevych, 2019) trained by BAAI. The dense retrievers are trained in an in-batch constrastive fashion using the loss function in Section B. The retriever was trained for 5 epochs using a batch size of 32 and a learning rate of 1e–5.

**Generator Training**: We have two types of generators: fine-tuning and few-shot. For fine-tuning, we train T5 and Code-T5 models. Each model was trained for 1 to 10 epochs with learning rates of 1e–4 and batch size of 16 to 64. The models were trained on an NVIDIA A6000 GPU for a total of approximately 24 hours. For few-shot learning, we used ChatGPT and GPT-4 in an incontext learning setup by providing three incontext examples and generation temperature of 0.01.

Following are further details of the metrics used for evaluation:

**BLEU**: For measuring the quality of the overall parse by the models, we EVALUATE the BLEU metric of the parse. We utilize the evaluate library of Huggingface and calculate the Sacrebleu (Post, 2018) metric for the NL2LTL and NL2CMD datasets. For the NL2Code dataset, we use the BLEU metric system and tokenization in the Docprompting paper. The formula for BLEU is as follows:

$$BLEU = bp.exp\left(\sum_{n=1}^{N} w_n \log p_n\right) \tag{2}$$

where $bp$ is the brevity penalty, $w_n$ is the weight of each n-gram precision and $p_n$ is the precision of each n-gram and $N$ is the total number of n-gram precisions considered in calculating BLEU.

**OVC metrics**: To measure ROLEX's ability to use expert knowledge to parse the OVCs, we exclusively calculate metrics related to the OVCs. This is done by first extracting the OVCs from the generated text via pattern matching with the OVCs in the gold lexicon. Then we calculate the unordered precision, recall and F1-score using the following formulas.

$$Precision = \frac{True\ Positive}{True\ Positive + False\ Positive} \tag{3}$$

$$Recall = \frac{True\ Positive}{True\ Positive + False\ Negative} \tag{4}$$

$$F1 - Score = 2 \times \frac{Precision \times Recall}{Precision + Recall} \tag{5}$$

Note that all OVCs in our evaluation sets are by definition and construction unseen during training and the evaluation metrics above focus only on the OVC performance. Thus all gains observed on these metrics will primarily reflect on the models' abilities to handle expert knowledge effectively during inference time and not affect any knowledge memorized during training.

**Gains**: ROLEX is designed to be an improved form of feedback system. In regular feedback systems for handling OVCs, an expert sits at the output helm, reads through each parse, and then provides corrective feedback for each erroneous OVC in the output. ROLEX reduces this amount of corrective feedback by reusing feedback from the expert. In this way, manual effort is reduced. We formulate an automatic way to measure this effort in the following manner: for each parse, the expert needs to spend 1 unit of work to read through the whole parse. Then the expert needs to spend 1 unit for each wrong OVC in the output. This number can easily be calculated by measuring the number of false negatives, i.e., OVCs missed in the output. The sum of these two numbers measures the total amount of human effort theoretically required to correct the OVCs. Hence the cost is:

$$Cost = Reading\_Cost + Error\_Cost \tag{6}$$

## F   Retriever Analysis

Table 7 shows the top-k recall on each benchmark dataset for five different set of retrievers: bm25 (Robertson et al., 2009), off-the-shelf small and large BGE sentence transformers from FlagEmbeddings (Xiao et al., 2023), and the fine-tuned versions of the aforementioned two models. We observe that bm25, being a lexical retriever, has lowest performance on all tasks. Off-the-shelf dense retrievers perform better, with the large variant doing better than the smaller one except in the case of NL2LTL dataset. However, the best performance is obtained with the fine-tuned retrievers. This indicates the necessity of retriever training for our task.

## G   Feedback Methods

Table 4 compares ROLEX using lexicons with two other methods: exemplar and ROLEX using docs.

| Dataset | Retriever | Downstream | | | |
| --- | --- | --- | --- | --- | --- |
| | | R@1 | R@5 | R@10 | R@20 |
| NL2LTL | bm25 | 1.92 | 9.13 | 12.50 | 17.79 |
| | Off-the-shelf | | | | |
| | -bge-small | 7.21 | 15.38 | 23.56 | 40.38 |
| | -bge-large | 6.25 | 17.79 | 24.52 | 36.54 |
| | Fine-tuned | | | | |
| | -bge-small | 11.54 | 30.77 | 43.75 | 47.12 |
| | -bge-large | 14.42 | 36.06 | 44.71 | 50.00 |
| NL2CMD | bm25 | 1.72 | 5.28 | 11.53 | 24.25 |
| | Off-the-shelf | | | | |
| | -bge-small | 40.19 | 57.97 | 63.25 | 68.64 |
| | -bge-small(FT) | 46.34 | 63.69 | 68.75 | 72.84 |
| | Fine-tuned | | | | |
| | -bge-large | 41.27 | 59.27 | 65.30 | 69.94 |
| | -bge-large(FT) | 52.59 | 68.86 | 72.20 | 75.32 |
| NL2Code | bm25 | 0.34 | 1.35 | 2.59 | 5.29 |
| | Off-the-shelf | | | | |
| | -bge-small | 9.57 | 16.22 | 17.91 | 19.59 |
| | -bge-small(FT) | 11.15 | 18.13 | 19.37 | 21.51 |
| | Fine-tuned | | | | |
| | -bge-large | 9.23 | 16.67 | 18.81 | 20.27 |
| | -bge-large(FT) | 11.37 | 18.69 | 20.38 | 22.18 |

Table 7: Top-k recall performance (R@k) for different sets of retrievers on each benchmark dataset.

**Examplar** refers to training the generator to use retrieved pairs of input-output as context during retrieval augmented generation. The idea comes from Pasupat et al. (2021) where the authors retrieved similar input-output pairs for semantic parsing. In our case, an off-the-shelf BGE large retriever is used to retrieve (specification, code) pairs from the training set of NL2Code to augment the training data with relevant context for training. Afterwards, during testing, the test input is augmented with these examples retrieved from the original training data.

The **ROLEX with docs** idea comes from the paper Zhou et al. (2023) where the models are conditioned to learn to use documents to overcome the OOD problem. For the NL2Code task in Table 4, we simply used the key of our lexicon as the document. The retriever used is the fine-tuned BGE large used in ROLEX + lexicons use-case. The generator is conditioned only upon NL $\oplus$ key.

## H   Gains Analysis

Table 8 shows the gains calculated as per our metric defined in Section 5.2 and and elaborated in Section E. For each dataset in the fine-tuned setting, we select the model with the highest difference in performance on OVCs. Then we calculate the reading cost and feedback cost. Reading cost is equal to the size of the dataset while feedback cost is equivalent to the false negative among the OVCs. This is then added to find the total. The percentage reduction between baseline and ROLEX is then calculated.We see 24.3% savings for NL2LTL, 5.6% for NL2Code, 11.9%for NL2CMD, and 32.9% for few-shot learning in NL2LTL. This shows the potential of ROLEX in reducing human effort.

## I   Error Analysis

Table 9 shows the detailed error analysis on the OVCs for the NL2LTL and NL2Code datasets. We selected the best models for each dataset, T5-Base and Code-T5 Base respectively, and analyzed the 25 most erroneous outputs. Four categories were identified as common errors: expert lexicon is available but not used, incorrect expert lexicon is used, expert lexicon is

| Dataset | Model | Reading | Error | Total | Reduction |
|---------|-------|---------|-------|-------|-----------|
| NL2LTL | CT5-B | 100 | 180 | 280 | |
| | +ROLEX | 100 | 112 | 212 | -24.3% |
| NL2Code | T5-B | 543 | 345 | 888 | |
| | +ROLEX | 543 | 295 | 838 | -5.6% |
| NL2CMD | CT5-S | 928 | 631 | 1559 | |
| | +ROLEX | 100 | 59 | 159 | -11.9% |
| Few-shot | GPT-4 | 100 | 137 | 237 | |
| | +ROLEX | 100 | 59 | 159 | -32.9% |

Table 8: Cost analysis of ROLEX compared to a regular feedback system using our defined "gain" metric. Each model is selected based on the maximum possible gain. This gain is shown as reduction in this table for each corresponding ROLEX.

available in the partial knowledge for retrieval but not retrieved, and OVC was hallucinated even though there were no corresponding expert lexicon in the input.

From the NL2LTL portion of the analysis, we observe that most errors corresponded to "hallucinating expert knowledge". This shows that the model tries to generalize without an expert lexicon, which is against our intended use case. The model also has a high probability of not using an expert lexicon or using an incorrect one.

For the NL2Code analysis, we omitted the "hallucinated EK" due to the large number of possible functions that the model can generate that may be an OVC. Also, due to the sparsity of OVCs in the dataset, we only select instances where an OVC is present. We observe that a large portion of the errors are when the model is not using an expert lexicon. This hints at the difficulty of using lexicons to resolve OVCs on this benchmark.

| Error Class | Percentage | |
|-------------|---------|---------|
| | NL2LTL | NL2Code |
| EK not used | 23.3 | 56.0 |
| Incorrect EK used | 23.3 | 16.0 |
| EK not retrieved | 6.7 | 28.0 |
| Hallucinated EK | 46.7 | - |

Table 9: Error analysis of ROLEX.

