# OpenReview forum: "Handling Open-Vocabulary Constructs in Formalizing Specifications: Retrieval Augmented Parsing with Expert Knowledge"
_colmweb.org/COLM/2024/Conference — COLM_

### Official Review · Reviewer_1xv4 · 2024-04-29

**Rating:** 6
**Confidence:** 5
**Ethics Flag:** 1

**Summary:**

The paper presents a study on the problem of Open-Vocabulary Constructs (OVCs) in the context of converting natural language (NL) specifications into formal languages. The authors propose a dynamic knowledge-augmented parsing (DKAP) problem, where a model receives input sentences and a dynamically growing expert knowledge lexicon that associates NL phrases with correct OVC constructs. The proposed approach, ROLEX, is a retrieval-augmented parsing method that uses this lexicon. The authors also introduce a new evaluation paradigm modeled after the DKAP problem and simulate the scenario across three formalization tasks: NL2LTL, NL2Code, and NL2CMD. The evaluations show that DKAP is a challenging problem, and ROLEX helps improve the performance of baseline models by effectively using dynamic expert knowledge.

**Questions To Authors:**

1.	The RAG part is not described in detail. How is the lexicon built? How big is it? Are all lexicons OOV? When retrieving, how to deal with the situation that contains multiple lexicons?

**Reasons To Accept:**

1.	Novel Approach to Dynamic Knowledge Augmentation: The paper presents a novel task setup called Dynamic Knowledge-Augmented Parsing (DKAP) to address the OVC problem. DKAP allows models to learn from and reuse dynamic, growing knowledge bases that record all knowledge provided by the user for OVCs encountered in previous sentences.

2.	Retrieval-Augmented Generation Model (ROLEX): The paper introduces a retrieval-augmented generation model called ROLEX, which collects small amounts of feedback from experts based on errors made by the parser when translating OVCs. This feedback is designed as a key-value lexicon, representing knowledge from the expert for the model to use in translating OVCs in future parses. The retriever module in ROLEX is trained to fetch relevant lexicons from the feedback database during each parse, while the generator module is conditioned to use this information when parsing. This approach helps improve the performance of baseline models by using dynamic expert knowledge effectively.

**Reasons To Reject:**

1.	Potential Overfitting: The paper mentions that the proposed method is trained to use the knowledge as it becomes available, without having to retrain every time a new construct is handled. However, this could potentially lead to overfitting, as the model might become too specialized in handling the constructs it has seen before, and perform poorly on new, unseen constructs. The paper could benefit from discussing this potential issue and proposing ways to mitigate it.

---

> ### Author Rebuttal · Authors · 2024-05-29
>
> We thank the reviewer for their insightful feedback.
>
> # Response to "Reasons To Reject":
>
> **Point 1**: We agree that the RAG model can potentially overfit to the retrieved constructs in the lexicon seen during training time. However, our experimental evidence shows this is not the case. Firstly, open-vocabulary constructs are by definition those not seen before. Our evaluations duly reflect this as gains are all seen using unseen constructs in feedback during inference. Secondly, the difference in performance between the validation and test sets is quite small indicating there is very little overfitting. For instance, for the Code-T5 base model trained via RoLex, the difference in BLEU performance between validation and test performances is a mere 0.2 BLEU score! For the NL2LTL domain, note that the training set is synthetic and has no overlap with the test set. The overall large gains in all of the settings indicate that the overfitting, if any, is small.
>
> # Response to "Questions to Authors":
>
> Some of these questions are addressed in the paper’s Appendix, which the reviewer may read for further clarification. To take each question in turn:
>
> * The lexicon is built using small snippets of text/documentation describing the construct.  For each evaluation testbed, the lexicons are constructed differently. Please see Appendix Section 8.3 for further details.
> * All of the lexicons in our testbeds are OOV compared to the training sets. This is accomplished by explicitly making sure that none of the test OVCs are seen in training examples.
> * Our retriever is designed to retrieve the top-k relevant lexicon entries (k=10 in our experiments). If a parse requires multiple lexicon entries, our fine-tuned retriever is trained to retrieve all of them within the top-k. Our generator is fine-tuned and conditioned upon the retrieval such that it identifies all of the relevant lexicon entries and uses them during generation. This is accomplished through specific training strategies during fine-tuning, such as teaching the models to learn to extract relevant lexicons first and then teaching them to do the semantic parsing task. These training strategies are highlighted in Section 4.3 of the paper.
>
> We will add discussions about reviewer's points in the final camera-ready version of the paper.

---

> > ### Author Response · Authors · 2024-06-04
> > **A gentle ping on our response to your feedback.**
> >
> > Dear reviewer,
> >
> > We thank you for your constructive feedback. We hope our clarifications above address your concerns. We are happy to answer any additional questions you may have.
> >
> > Thanks for your time in helping us improve the paper.

---

### Official Review · Reviewer_2rdn · 2024-05-01

**Rating:** 7
**Confidence:** 2
**Ethics Flag:** 1

**Summary:**

The paper presents an approach to handling open-vocabulary constructs in semantic parsing, based on gathering expert feedback provided at inferene time. In particular, the paper presents (i) a semantic parsing setting called dynamic knowledge-augmented parsing, where said expert feedback needs to be taken into account, and (ii) a retrieval-augmented generation model to tackle the task via a key-value lexicon where keys are descriptions of OVCs and values are the desired responses. Experiments on a range of tasks show that the proposed approach provides good performance.

**Questions To Authors:**

A few typos:

p.1: an linear -> a linear

p.3: problem,s -> problems,

p.9: the for OVC errors for -> extra "for"

**Reasons To Accept:**

The paper addresses an interesting practical problem, that of open-vocabulary constructs in semantic parsing.

The paper is clear, well-structured and easy to follow; and has sufficient substance.

The proposed approach provides good empirical results across a range of tasks.

The paper features detailed analysis of the results, including ablations in a component by component basis.

**Reasons To Reject:**

The problem is somewhat niche - one does not often have high-quality expert feedback at inference time.

---

> ### Author Rebuttal · Authors · 2024-05-29
>
> We thank the reviewer for their constructive feedback.
>
> # Response to "Reasons to Reject":
>
> **Point 1**: We agree that the problem setting is not typical in open-domain NLP problems, including standard semantic parsing application areas such as Text-to-SQL or Knowledge-based QA.
>
> However, as we argue in the paper (beginning paragraph of Section 3), the setting we introduce is especially well-suited for formal verification, where there are a large number of problem domains in which translating text specifications to formal statements is a serious burden preventing progress in verification and slowing the development of protocol implementations. For example, there are thousands of RFCs out there and, for each of them, the expert needs to parse the specification to formal language manually. Also, this project and associated datasets were developed from collaborations with formal verification researchers who want to verify Network File Systems but are unable to do so due to the difficulty of parsing specifications into formal models. Hence, their difficulty further necessitates the need for our solution.
>
> We will add text to the paper to better highlight this motivation and the broader potential for impact in the final version.
>
> # Response to "Questions to Author":
>
> We thank the reviewers for pointing out these typos/errors. We will fix them in the final camera-ready version of the paper.

---

> > ### Comment · Reviewer_2rdn · 2024-05-31
> > **Thanks for the clarification**
> >
> > Thank you for the clarification. I do think making this more explicit in the final version, if accepted, should make the motivation of the paper clearer.
> >
> > I am not updating my score at this point, since it was already quite positive.

---

> > > ### Author Response · Authors · 2024-06-01
> > > **Thank you.**
> > >
> > > Thanks for your constructive suggestions. We will add these to final version.

---

### Official Review · Reviewer_wWrJ · 2024-05-14

**Rating:** 6
**Confidence:** 4
**Ethics Flag:** 1

**Summary:**

This paper presents a method to insert open-vocabulary constructs i.e. constructs such as names of methods/functions at inference time to a LLM during generation. The goal is to do a more controllable generation where practitioners can correct mistakes or change behavior (e.g. asking it to use np.linalg.solve when the default gen is scipy.linalg.solve). The paper advocates storing these feedback/expert knowledge in a key-value store and trains a retriever to retrieve from the store and a generator that uses these retrievals to produce generations with the desired behavior. The memory grows as an expert keeps updating the memory and the retriever can retrieve from the growing knowledge. The basic idea being that an error once fixed because of edits will not be made again as the edits can be retrieved from the memory. The retriever is based on standard sentence-transformers (BGE) and for generators T5-based models as well as ChatGPT is used.

The paper evaluates on three semantic parsing/program synthesis tasks and shows impressive performance gains with the retrieval from feedback memory.

**Reasons To Accept:**

1. The paper tackles a practical problem of generalizing your RAG model to unseen constructs and presents a practical and usable solution.
2. The paper is written clearly and was easy to understand
3. The proposed solution works well.

**Reasons To Reject:**

1. Novelty: The proposed solution is not novel and this approach is known to the community. Having a memory of feedback and using them with a RAG-based approach is a nice solution but it has been explored before (e.g. Mishra et al 2022 and Das et al 2021). It would be nice if the author clearly differentiates their work from other existing work.
2. The paper is motivated with a human-in-the-loop set up but the experiments are done by deriving data synthetically. While it was enough to show that the idea works but it would have been nice to do some actual human-in-the-loop experiments. I am guessing there are other practical considerations which gets left out when the experiments are conducted in a simulated scenario.
3. The paper has considered fine-tuning the retriever on the knowledge stored in the feedback memory. Do you have a sense if the retriever become specialized for this domain and loses it general ability to retrieve from general documents? In other words, does fine-tuning in general have any adverse effects on generalizability?


References
* Das et al 2021: Case-Based Reasoning for Natural Language Queries over Knowledge Bases (refer to Sec 3.4)
* Mishra et al 2022: Towards Teachable Reasoning Systems: Using a Dynamic Memory of User Feedback for Continual System Improvement

---

> ### Author Rebuttal · Authors · 2024-05-29
>
> We thank the reviewer for their constructive feedback and related work suggestions.
>
> # Reponse to "Reasons to Reject":
>
> **Point 1**: Approaches in referred papers have similarities (using RAG to model feedback) but differs in tasks and training methods.
> * (Mishra et. al 2022) , while using the concept of dynamically growing feedback database, differs in downstream domain task (QA vs Semantic Parsing). They also utilize free-form feedback compared to our lexicon-based feedback, which is more efficient for addressing constructs ( Table 4(b) ). We also train our retrievers for domain adaptation, compared to using off-the-shelf retrievers like them, due to the non-generalizability of our domains. Secondly, they use a trick called ``forced generation’’ which generates multiple inputs with each retrieved feedback using off-the-shelf generators.  We propose different unique training schemes that teach the model to use feedback effectively (Section 4.2 for details).
> * (Das et. al 2021) uses training examples/cases as feedback and have a static database of such examples. Whereas, our feedback is designed as a dynamic database of lexicons growing with each inference parse.
>
>
>
> **Point 2**:  We agree that actual HIL experiments can showcase practical considerations ( HCI aspects etc.). However, semantic parsing problems we consider require domain knowledge and trained users. Hence, testing using such expert users can be difficult and expensive. This is especially true for our NL2LTL setting, in Network File Systems domain, where the target formalism is niche and our test set required many iterations and months of input from formal verification experts. Hence, we settled for the middle ground of using simulated environments common across the literature.
>
> We assume “deriving data synthetically” refers to how human feedback data was simulated and not test datasets themselves, as all of them are real-world datasets.
>
> **Point 3**: Fine-tuning retrievers can affect generalizability. However, our test domains are specialized and models require domain-specific knowledge. Hence, fine-tuning, compared to off-the-shelf RAG, yields significant performance gains, as shown in results in Table 3(b) and 7.  Building generalized retrievers that can work across domains (e.g. from NL2LTL to the NL2Code domains) can be interesting future work in this space.
>
> In the final camera-ready version of the paper, we will add discussions to these points.

---

> > ### Comment · Reviewer_wWrJ · 2024-05-31
> > **Thank you for the comment**
> >
> > Thank you for the response. Please address them in the paper as well and try to clearly differentiate from the aforementioned related work. Also I revisited Das et al 2021 recently, and I am not sure that their experiment is in a static setting.
> >
> > I have decided to keep my scores the same. Thank you.

---

> > > ### Author Response · Authors · 2024-06-01
> > >
> > > In Das et. al 2021, the general methodology proposed in Section 2.2 (specifically in “Details of Input”) proposes their RAG solution in a static database setting. However, in Section 3.4.2, they motivate a HIL setup where they acquire example cases for unseen entities to improve upon the OOD test cases only. These examples are obtained from another dataset (SimpleQuestions) and users with knowledge about the KB. Their main motivation is to show that their method can improve performance without retraining by just expanding the database with relevant examples. We could not add these details in the main response due to the character limit. However note that their method and evaluation are still significantly different from ours:
> > > * Our whole setup and evaluation assume a dynamic memory that grows from zero as new out-of-vocabulary constructs are seen and corrected with streams of test inferences to address the out-of-vocabulary scenario explicitly. This is unlike Das et. al 2021 who did an added HIL experiment to show that their idea can be expanded to dynamic memory.
> > > * Their HIL feedback requires writing whole train cases whereas ours requires the expert to add a small idiomatic description and the construct value as a lexicon. This is more efficient and effective in our use case, as shown in our results in Table 4(b).
> > > * Das et. al 2021 generator incorporates a Kullback–Leibler loss function to reduce the difference between when cases are provided and when they are not. This loss, while suitable for their setting, is inappropriate for ours. This is because, in our case, we want the target formalism to explicitly use the open-vocabulary constructs defined in the lexicon feedback, even if it means that there is a significant deviation from baseline (models which do not use feedback). Hence, we formulate training schemes that teach the model to use feedback/knowledge more effectively (detailed in Section 4.2 of our paper).
> > > * Das et. al 2021 also propose a revise step when modeling to specifically address the issue of domain mismatch when parsing NL to LF. They do this by aligning generated relations with relations present in the local neighborhood of the query entity in the KG, discovered via proximity between sentence embeddings. Hence, this modeling step is only suitable for their problem setting due to the nature of knowledge graphs. It is not ideal for our setting.
> > >
> > > We thank the reviewer for pointing out the HIL setting, present in Section 3.4.2, in Das et. al 2021. We also thank them for engaging in this conversation which helped make our paper stronger.

---

> > > > ### Author Response · Authors · 2024-06-04
> > > > **A gentle ping on the above clarification with respect to Das et al., 2021**
> > > >
> > > > Dear reviewer
> > > >
> > > > We were wondering if you had a chance to look at the clarifications regarding Das et al. 2021.
> > > >
> > > > We believe our work shares similarities with the experiment in Das et al., 2021, but our work mainly focuses on OOV problem and addresses the important distinctions that arise in the problem settings we consider. As mentioned above we will add these discussions to the related work in the final version. We hope this addresses your concern in this regard. Thanks again for your constructive feedback on our work.

---

> > > > > ### Comment · Reviewer_wWrJ · 2024-06-04
> > > > > **Thank you for the detailed comparison**
> > > > >
> > > > > Dear authors,
> > > > >
> > > > > Thank you for the detailed comparison. Yes, this makes sense and I understand the differences clearly. Please update the paper accordingly. I have updated my score as well.

---

### Decision · Program_Chairs · 2024-07-10

**Decision:**

Accept

**Comment:**

The authors examine parsing problems where the parser must utilize constructs not known at training time, such as the use of API not know during training. They assume access to user feedback in the form of a lexicon (e.g., NL expresssion -> function name), which accumulates over time from previous model iterations and is accessed in a RAG / non-parametric manner.

The apporach, though not ground-breaking, is is sufficiently novel, even if the problem is somewhat niche, as acknowledge by reviewer. The paper is well-written, with no concerns about the evaluation or reliability of the findings. Several suggestions for improvement are provided, including better positioning with respect to previous non-parametric metrics, such as the work by Das et al. (2021), which also considers an online/dynamic setup and extending the discussion of RAG.